# Virus-like Particle Vaccine for Feline Panleukopenia: Immunogenicity and Protective Efficacy in Cats

**DOI:** 10.3390/vaccines13070684

**Published:** 2025-06-25

**Authors:** Tongyan Wang, Hongchao Wu, Yanwei Wang, Yang Guan, Yujiao Cao, Lulu Wang, Mengyue Wang, Feifei Tan, Wenqiang Pang, Kegong Tian

**Affiliations:** 1College of Veterinary Medicine, Henan Agricultural University, Zhengzhou 450046, China; wangtongyan@pulike.com.cn (T.W.); wuhongchao@pulike.com.cn (H.W.); 2National Research Center for Veterinary Medicine, Road Cuiwei, High-Tech District, Luoyang 471003, China; wangyanwei@pulike.com.cn (Y.W.); guanyang@pulike.com.cn (Y.G.); caoyujiao@pulike.com.cn (Y.C.); wanglulu@pulike.com.cn (L.W.); wangmengyue@pulike.com.cn (M.W.); tanfeifei@pulike.com.cn (F.T.); 3Pulike Biological Engineering, Inc., Luoyang 471000, China

**Keywords:** feline parvovirus, virus-like particles (VLPs), vaccine

## Abstract

Background/Objectives: Feline panleukopenia, caused by FPV, is a highly contagious disease in cats. Current vaccines face challenges including complex production, high cost, and safety risks. Developing safer, more efficient alternatives is crucial. This study aimed to produce FPV virus-like particles (VLPs) using a recombinant baculovirus system expressing the VP2 gene and evaluate their immunogenicity and protective efficacy in cats. Methods: Sf9 insect cells were infected with recombinant baculovirus to express VP2 protein. The VP2 protein was purified using ultrafiltration and size-exclusion chromatography (SEC). Dynamic light scattering (DLS) and transmission electron microscopy (TEM) confirmed the assembly of VLPs. Twenty healthy cats were randomly divided into four groups; three groups received different doses (5 μg, 15 μg, and 45 μg) of FPV VLP vaccine, while the fourth group served as the control group immunized with PBS. Blood samples were collected on day 21 to measure hemagglutination inhibition (HI) and virus-neutralizing (VN) antibody responses. Cats in the 15 μg dose group were challenged with virulent FPV strain 708 on day 21, and clinical signs and white blood cell counts were monitored for 10 days. Results: Immunized cats exhibited significantly higher HI and VN antibody titers compared to controls. After challenge, vaccinated cats showed no clinical signs of disease, and their white blood cell counts remained stable. In contrast, control cats developed severe symptoms and experienced significant leukopenia. Conclusions: The FPV VLP vaccine generated in this study are highly immunogenic and provide effective protection against virulent FPV challenge, demonstrating their potential as a safer vaccine candidate for feline panleukopenia.

## 1. Introduction

Feline panleukopenia, caused by feline parvovirus (FPV), is a highly contagious and often fatal disease that poses a major threat to feline health [1]. The clinical presentation typically includes fever, persistent vomiting, diarrhea, dehydration, a significant decrease in white blood cell count (leukopenia), and hemorrhagic enteritis. The mortality rate associated with the disease varies depending on the severity of clinical signs, ranging from 25% to as high as 100%. While FPV can affect cats of all ages, kittens are particularly susceptible, with mortality rates exceeding 90% in this age group [1,2]. FPV was first identified in domestic cats in the 1920s. Since then, it has been isolated from a wide range of species, including primates, raccoons, and various wild and captive carnivores [3,4,5], highlighting its broad host range.

Among the many strategies for disease prevention and control, vaccination is the most economical and effective strategy for disease prevention and control. The currently licensed FPV vaccines include inactivated vaccines and live attenuated vaccines. The production processes of inactivated and live attenuated vaccines generally involve cultivating large quantities of live viruses, followed by physical or chemical inactivation for inactivated vaccines, or attenuation treatment for live attenuated vaccines. Because the production of live vaccines requires large-scale propagation of viruses, these processes are time-consuming and must be carried out in high-level biosafety facilities to prevent potential infection risks [6].

Thanks to advances in genetic engineering, virus-like particles (VLPs) have gained recognition as a promising and secure platform for vaccine development against viral infections. VLPs are nanostructures that mimic the size and shape of viruses, adopting icosahedral configurations. These structures are formed by the spontaneous assembly of multiple copies of one or more recombinant viral structural proteins, without including any viral genetic material [7]. Because of their highly organized and repetitive presentation of antigenic determinants, VLPs effectively stimulate strong antibody-mediated immune responses and potent cellular immune responses mediated by T cells [8].

FPV (currently classified as Protoparvovirus carnivoran 1) belongs to the family Parvoviridae, subfamily Parvovirinae, genus Protoparvovirus [9], and has a symmetrical icosahedral structure [10]. Its genome consists of a linear, single-stranded DNA (ssDNA) molecule approximately 5 kb in length, containing two major open reading frames (ORFs). The first open reading frame (initiating at the 5′ end) encodes the non-structural proteins, such as the NS1 protein, which are involved in viral replication. The second open reading frame encodes the structural proteins, including the major capsid proteins VP1 and VP2, which are crucial for the formation of viral particles and host cell infection. VP1 contains the entire sequence of VP2, with an additional N-terminal region of about 140 amino acids. VP2 is the major capsid protein of the virus. During viral maturation, the VP2 protein can be partially cleaved by host proteases to generate the VP3 protein [4]. The structural protein VP2 is the major capsid protein that determines the antigenicity, pathogenicity, and host infection spectrum of the pathogen [11], with major epitopes that stimulate the production of neutralizing antibodies and serve as the target antigen for subunit vaccines [12]. The VP2 proteins of canine parvovirus (CPV) and mink enteritis virus (MEV) of family Parvoviridae are capable of self-assembling into virus-like particles [13,14].

In light of the growing global trend of pet ownership and the increasing demand for safe and effective vaccines, the objective of this study was developing a recombinant baculovirus that expresses FPV VP2 and resembles VLPs, and to evaluate the immunogenicity of the FPV VLPs in cats.

## 2. Materials and Methods

### 2.1. Virus and Cell Line

Feline kidney F81 cells were cultured at 37 °C in a 5% CO_2_ atmosphere in RPMI 1640 medium. This medium was supplemented with 8% fetal bovine serum (FBS, PAN) containing 100 units/mL penicillin and 100 μg/mL streptomycin. Both the cultivation of the virulent FPV strain 708 and the serum neutralization test were conducted in F81 cells.

Sf9 insect cells were maintained in suspension in serum-free SF900 Ⅱ medium (Gibco, Grand Island, NY, USA) at 27 ± 0.5 °C in spinner flasks at 90~120 rpm. A recombinant baculovirus based on the Autographa California multiple enveloped nuclear polyhedrosisvirus (AcMNPV) was propagated, and FPV VP2 protein expression was carried out in the Sf9 cells.

### 2.2. Animals

The subjects, 20-week-old Chinese rural cats, including both males and females, free of FPV, feline calicivirus (FCV), feline herpesvirus 1 (FHV-1), feline coronavirus (FCoV) and FPV antibody was less than 1:4, were obtained from a holding facility in Luoyang and were used to evaluate the efficacy of the VLP-based vaccine candidate. The cats were housed under feline panleukopenia virus (FPV)-free conditions throughout the study. All animal samples were collected in accordance with the guidelines approved by the Animal Care and Ethics Committee of the National Research Center for Veterinary Medicine (Permit No. 20200725023).

### 2.3. Antisera

A polyclonal antiserum against FPV was generated by immunizing two 2 kg New Zealand White rabbits with a twice-inactivated FPV vaccine, at a three-week interval between immunizations. Blood samples were collected from the animals on day 28 following the booster immunization, and serum was isolated and stored at −20 °C for use in subsequent experiments.

### 2.4. Generation of Recombinant Baculovirus

The codon usage of the VP2 gene (GenBank No. MH329286.1) was optimized to align with the codon preference of Sf9 insect cells. For the construction of the recombinant donor plasmid, two copies of the VP2 gene were inserted into the SmaI/SphI and BamHI/HindIII restriction sites of the baculovirus transfer vector pFastBac™ Dual-Hr3. The homologous region 3 (Hr3) was positioned between the pH and p10 promoters to enhance expression efficiency as previously described [15]. The resulting donor plasmid was transformed into E. coli DH10Bac competent cells containing the AcMNPV bacmid (Invitrogen, Carlsbad, CA, USA), allowing the transposition of the VP2 gene into the bacmid genome to generate a recombinant bacmid. To rescue the recombinant baculovirus, Sf9 cells were seeded at a density of 1 × 10^6^ cells per well in 6-well plates and transfected with 4 μg of purified bacmid DNA using Cellfectin^®^ II reagent (Invitrogen), following the manufacturer’s protocol. Following three serial passages, the virus titer in the harvested supernatant was determined by plaque assay.

### 2.5. Identified FPV VP2 Protein Expression

VP2 expression was detected using an indirect fluorescence assay (IFA). Briefly, Sf9 cells were infected with a recombinant baculovirus expressing VP2 proteins, and fixed after 72 h. A rabbit polyclonal antiserum against FPV was used as the primary antibody at a dilution of 1:500, followed by a FITC-labeled goat anti-rabbit IgG (Invitrogen) as the secondary antibody at a dilution of 1:5000. Specific fluorescence was observed under a fluorescence microscope (Leica, Wetzlar, Germany) to determine the expression of the VP2 protein.

Sf9 cells were infected with a recombinant baculovirus at a multiplicity of infection (MOI) of 1PFU per cell. The cells were harvested 72 h post-infection, washed with PBS, and lysed in a 25 mM bicarbonate solution at 4 °C for 90 min. Cell debris was removed by low-speed centrifugation. The supernatant of cell lysate was subjected to 12% SDS-PAGE gel electrophoresis, followed by Coomassie Brilliant Blue staining and Western blot analysis. For Western blotting, the separated proteins were transferred onto a nitrocellulose (NC) membrane (BBI). A rabbit polyclonal antiserum against FPV was used as the primary antibody at a dilution of 1:500. Horseradish peroxidase (HRP)-labeled goat anti-rabbit IgG (Invitrogen) was used as the secondary antibody at a dilution of 1:2000. After washing with PBST, the membrane was incubated with DAB reagent (Beyotime, Shanghai, China) for visualization.

Further characterization was performed using a hemagglutination (HA) test at 4 °C, with a 1% (*v*/*v* in PBS) suspension of pig erythrocytes, as previously described [16]. Briefly, the sample was serially two-fold diluted in phosphate-buffered saline (PBS, pH 6.5) in a 96-well V-bottom microtiter plate. An equal volume of 1% (*v*/*v*) pig erythrocytes was added to each well. After gentle mixing, the plate was incubated at 4 °C for 90 min. Hemagglutination was defined as complete agglutination of red blood cells, forming a visible lattice across the well surface. The HA titer was recorded as the reciprocal of the highest dilution showing complete agglutination.

### 2.6. Purification and Characterization of VLPs

The expression and pretreatment of the VP2 protein were performed as described in Section 2.5. The cell lysate supernatant was concentrated using an Ultra Filtration Concentrator Tube and applied to a pre-equilibrated gel filtration column HiLoad 16/600 Superdex 200-pg (GE Healthcare, Chicago, IL, USA) for preparative size exclusion chromatography. Peaks corresponding to FPV VP2 were identified by 12% sodium dodecyl sulfate-polyacrylamide gel electrophoresis (SDS-PAGE) of the elution fractions.

Size exclusion chromatography (SEC) was performed using high-performance liquid chromatography (HPLC) (Shimadzu, Kyoto, Japan) with a size-exclusion chromatography column (Sepax, SRT SEC-500 PN: 215500-7830, 7.8 × 300 mm, 5 µm) and UV (280 nm) detection. Dynamic light scattering (DLS) experiments were conducted using a DynaPro Light Scattering system (Malvern Panalytical, NANO ZSE) (Wyatt Technology Europe GmbH, Dernbach, Germany). The samples were analyzed at a protein concentration yielding an absorbance of 1.0 at 280 nm. The buffer used in the measurements contained 250 mM sodium phosphate (pH 7.4)

Purified VLPs were placed onto copper grids and allowed to adhere for 5 min at ambient temperature. Following adsorption, the grids were carefully blotted dry with filter paper and then stained with a 3% phosphotungstic acid (PTA) solution for 5 min. Any surplus liquid was removed by touching the edge of the grid with filter paper. The samples were subsequently analyzed using a transmission electron microscope (TEM) at an accelerating voltage of 80 kV (HITACHI HT-7700).

### 2.7. The Immunity of FPV VLPs

Under aseptic conditions, the purified VLPs (45, 15, and 5 μg per 1.0 mL) were formulated with SEPPIC gel adjuvant and stored at 2–8 °C. Twenty cats were evenly distributed into four experimental groups (as detailed in Table 1) and immunized by the subcutaneous route. Groups 1, 2, and 3 were immunized with a different VLP vaccine at day 0. Group 4 was inoculated with PBS as the unvaccinated control group. Blood samples were collected at 21 days post-immunization to detect the hemagglutination inhibition (HI) antibody and virus-neutralizing (VN) antibody.

The cats in group 2 and the cats in group 4 were challenged with 1 × 10^5^ 50% fluorescent antibody infective dose (FAID_50_) per 1.0 mL of the virulent FPV strain 708 on day 21 post-immunization. The challenged cats were monitored daily for 10 days.

Anticoagulated blood samples were collected from the cephalic or jugular vein of each cat before challenge and on the 6th day post-challenge. White blood cell (WBC) counts were subsequently determined using a Veterinary Automatic Hematology Analyzer (Mindray, Inc., Shenzhen, China), according to the manufacturer’s instructions. The pre-challenge WBC count was used as the baseline value for each cat. To assess changes in leukocyte levels, the WBC count on day 6 post-challenge was normalized to the corresponding baseline value, and the resulting ratio was defined as the leukocyte ratio.

Intestinal tissues were harvested 10 days after the FPV challenge, fixed in 4% paraformaldehyde, embedded in paraffin wax, and sectioned prior to hematoxylin and eosin staining (H&E). Monoclonal antibody 4B1 (Luoyang Putai Biotechnology Co., Ltd., Luoyang, China) against FPV VP2 and goat anti-mouse IgG conjugated to HRP (Thermo Fisher Scientific, Waltham, MA USA) were used for immunohistochemical (IHC) assays. Stained sections were visualized using a Leica DM2500 microscope (Leica Microsystems).

### 2.8. Serological Assays

The serum was heat-inactivated at 56 °C for 30 min. The HI antibody titers were analyzed using the previous method [13]. Briefly, a serial two-fold dilution of inactivated serum samples was mixed with 8 haemagglutination (HA) units of FPV in a 1:1 volume ratio. After incubation at 37 °C for 30 min, 1% (*v*/*v*) pig erythrocytes were added, and the samples were then incubated at 4 °C for a further 90 min. The titer was calculated as the highest dilution at which agglutination was still observed.

The serum was analyzed for FPV neutralizing antibody titers by heat-inactivating it at 56 °C for 30 min, followed by a two-fold serial dilution ranging from 1:2 to 1:2048 according to the previous method [13]. A 200 FAID_50_ dose of FPV was mixed with the serum sample at a volume ratio of 1:1 and incubated at 37 °C for 1 h. Then, F81 cells (2 × 10^4^ cells/100 μL) were added. After a 5-day incubation at 37 °C with 5% CO_2_, antibody titers were measured by IFA and expressed as the reciprocal of the highest dilution at which infection of the F81 cells was inhibited in 50% of the culture wells.

### 2.9. Statistics

Prior to conducting group comparisons, Bartlett’s test was performed to assess the assumption of homogeneity of variances across the hemagglutination inhibition (HI) antibody titers, neutralizing antibody titers, and white blood cell count changes. The results indicated that all *p*-values were greater than 0.5, suggesting no significant differences in variances among the groups. Therefore, pairwise comparisons were carried out using independent samples *t*-tests.

Data analysis was conducted using Python (version 3.12.9) with the SciPy package (version 1.13.1). A one-tailed independent *t*-test was employed to compare each immunization group with its corresponding control group. Fisher’s exact test was utilized to evaluate whether there are statistically significant differences in various clinical symptoms across different groups.

## 3. Results

### 3.1. Identified FPV VP2 Protein Expression

Sf9 cells were seeded in a 6-well plate at a density of 1 × 10^6^ cells per well and transfected with 4 μg of recombinant bacmid using Cellfectin II Transfection Reagent to generate recombinant baculovirus according to the manufacturer’s instructions. After two passages, Sf9 cells infected with the virus exhibited typical cytopathic effects (CPE), including increased cell diameter, oration of intracellular vesicles, and detachment from the culture surface (Figure 1A). In contrast, no morphological changes were observed in the Sf9 control cells cultured under identical conditions (Figure 1B), indicating successful rescue of the recombinant baculovirus.

Sf9 cells were infected with a recombinant baculovirus at a multiplicity of infection (MOI) of 1 plaque-forming unit (PFU) per cell. At 72 h post-infection, VP2 protein expression was confirmed by indirect fluorescence assay (IFA). Specific green fluorescence was observed in cells infected with recombinant baculovirus (Figure 1C), whereas no signal was detected in the uninfected Sf9 control cell (Figure 1D).

The expression of VP2 was further analyzed by 12% SDS-PAGE and Western blot (Figure 1E and Figure 1F, respectively, lane 3). In both assays, VP2 appeared as a band with an apparent molecular weight of approximately 65 kDa, which is consistent with its predicted molecular weight of 64.68 kDa. These results indicate successful expression of the FPV VP2 protein in insect cells.

Moreover, hemagglutination (HA) activity was detected in the VP2-expressing insect cells (Figure 1G), whereas Sf9 control cells were negative for HA activity (Figure 1H), suggesting that the recombinant VP2 protein retained functional properties similar to those of the native viral protein.

### 3.2. Purification and Characterization of VLPs

Sf9 cells were infected with recombinant baculovirus for 72 h post-infection(hpi). The cell lysate supernatant was concentrated using Ultra Filtration Concentrator Tubes and further purified using a gel filtration column (HiLoad 16/600 Superdex 200-pg). The elution peak corresponding to the FPV VP2 protein was observed at a retention volume of approximately 48.8 mL (Figure 2A), as confirmed by SDS-PAGE. The purity of the final recombinant VP2 protein was assessed by SDS-PAGE and estimated to be around 85% using Quantity One software (version 5.2.1) (Bio-Rad, Hercules, CA, USA), as shown in Figure 2B.

Size-exclusion chromatography (SEC) analysis was performed using a high-performance liquid chromatography (HPLC) system (Shimadzu LC-2050C) equipped with a size-exclusion chromatography column. Elution was carried out under isocratic conditions at a flow rate of 0.5 mL/min for 13.5 min, with phosphate-buffered saline (PBS) serving as the mobile phase. The main peak observed in the chromatogram corresponded to the virus-like particle (VLP) nanoparticles (Figure 2C).

Dynamic light scattering (DLS) analysis demonstrated that nearly 100% of the purified VP2 protein assembled into a homogeneous high-molecular-weight complex, indicating the formation of virus-like particles (VLPs) (Figure 2D). To further characterize the structure of these VLPs, transmission electron microscopy (TEM) was performed. The TEM images revealed that the in vitro assembled FPV VLPs exhibited a spherical morphology with an average diameter of approximately 25nm, consistent with the DLS measurements (Figure 2E).

### 3.3. The Immunogenicity of the FPV VLPs

Twenty cats were evenly distributed into four experimental groups. Group 1–3 cats were immunized with FPV VLP vaccine at varying doses, and no adverse effects were observed in any of the experimental groups following vaccination. These findings indicate that the FPV VLP vaccine is safe and well-tolerated across the tested dose range in feline subjects. At 21 days post-immunization, all immunized cats with three VLP vaccines containing different antigen concentrations exhibited high levels of both HI antibodies (Figure 3A) and VN antibodies (Figure 3B), which were significantly higher than those in the PBS control group. Furthermore, the HI and VN antibody titers at day 21 post-immunization demonstrated a clear dose-dependent trend. Among them, the 45 μg VLP vaccine group had mean HI antibody titers of 11log_2_ and VN antibody titers of 9.2log_2_, indicating a strong immunogenic response.

The protective efficacy of the FPV VLP vaccine containing 15 μg of VLPs was further evaluated by challenging cats with the FPV strain 708. After challenge, cats in the PBS control group exhibited typical symptoms of feline panleukopenia (Table 2, shown in Appendix A), such as vomiting (2/5 cats, Figure 4A), diarrhea (5/5 cats, Figure 4B), and fever (3/5 cats, Figure 4C). Moreover, in the PBS control group, rectal temperatures of two cats exceeded 40 °C, and three cats died (Figure 4D). In contrast, no symptoms of feline panleukopenia were observed in any of the vaccinated cats.

Anticoagulated blood samples were collected from the cephalic or jugular vein of each cat before challenge and on the 6th day post-challenge. The pre-challenge WBC count was used as the baseline value for each cat. As shown in Figure 4E, the WBC counts of cats in the PBS control group exhibited a significant decrease at 6 days post-challenge, reaching 10% or less of their baseline values. In contrast, the VLP vaccine-immunized group did not show a marked decline in WBC counts; rather, all individual counts remained at least 80% above baseline values.

Following necropsy at 10 days post-infection or time of death, intestinal tissues were subjected to histopathological examination. As depicted in Figure 4F, cats in the PBS control group demonstrated villous sloughing off, lamina propria hemorrhage, and necrotic lesions, with FPV antigen localization confirmed by IHC. Conversely, intestinal sections from VLP vaccine vaccinated cats showed no detectable pathological alterations, and IHC staining was negative for FPV.

## 4. Discussion

Virus-like particles (VLPs) represent a promising next-generation vaccine strategy. Structurally similar to native viruses, VLPs are non-infectious due to the absence of viral genetic material, eliminating the risks associated with viral replication and pathogenicity. Their particulate nature and repetitive antigenic structure make them highly effective at stimulating both humoral and cellular immune responses. The advantages of VLP-based vaccines include their strong safety record, high immunogenicity, and adaptability for use in multivalent or cross-protective vaccine designs. Several VLP-based vaccines have already been licensed and successfully implemented in clinical settings, including those targeting human papillomavirus (HPV) and porcine circovirus type 2 (PCV_2_). These examples underscore the practicality and effectiveness of this platform in real-world applications.

The baculovirus expression vector system (BEVS) is a powerful tool for the production of recombinant eukaryotic proteins, particularly those requiring post-translational modifications such as glycosylation and phosphorylation. Compared to other commonly used systems—such as bacterial, yeast, and mammalian expression platforms—BEVS offers several key advantages in cost, speed, and ease of use. Insect cell cultures (e.g., Sf9 or Sf21) can be grown at high densities in serum-free media, significantly reducing culture costs compared to mammalian systems. Additionally, the infrastructure required is less complex, making BEVS more accessible and cost-effective. Protein expression typically reaches high levels within 48–72 h after infection, which is faster than many mammalian systems. The system also allows easy optimization through adjustments in multiplicity of infection (MOI), temperature, and harvest time. BEVS is highly flexible and user-friendly, with a wide range of commercial vectors and host cell lines available. It supports the expression of complex proteins, including membrane proteins, virus-like particles (VLPs), and multi-subunit complexes. Moreover, baculoviruses are non-pathogenic to vertebrates, classified under BSL-1, making them safe for routine laboratory use. These features have made the baculovirus expression vector system a popular choice for the production of virus-like particles (VLPs) [17,18,19,20].

Previous studies have demonstrated that Hr3 enhances the expression of target genes by modulating enhancer activity [21,22]. In this study, FPV VP2 protein was expressed using the baculovirus expression system with Hr3. The FPV VP2 protein expressed using a recombinant baculovirus system yielded 0.63 mg/L (estimation of protein expression levels based on quantitation of purified protein), with a hemagglutination (HA) titer of 19log_2_, markedly surpassing the HA titer of CPV VP2 protein expressed in silkworm pupae, which was only 9log_2_ [13]. Western blot analysis of the recombinant FPV VP2 protein revealed not only the expected 65 kDa specific band but also several non-specific bands, suggesting possible cross-reactivity with host cell proteins present in the polyclonal antiserum. In addition, the recombinant baculovirus infected Sf9 cells in suspension culture with a particularly high HA titer (19log_2_) suitable for large-scale production.

In this study, the VP2 protein expressed by the recombinant baculovirus was purified and characterized using high-performance liquid chromatography with size-exclusion chromatography (HPLC-SEC), dynamic light scattering (DLS), and transmission electron microscopy (TEM). The results confirmed the formation of virus-like particles (VLPs), whose size and morphology were consistent with those of the native virus and VLPs reported in previous studies [23]. However, an additional peak was detected at around 20–21 min during HPLC-SEC analysis, suggesting that the purified protein may still contain some impurities, likely attributable to co-purified host cell proteins or other contaminating substances.

In vaccine research, the application of adjuvants plays a crucial role in enhancing vaccine immunogenicity and protective efficacy. As a commonly used adjuvant, the gel adjuvant has been extensively studied in vaccine development. For instance, Jiao et al. [23] utilized gel adjuvant to prepare bacterium-like particle vaccines and evaluated their immunogenicity. Therefore, this study also used gel adjuvant to prepare feline parvovirus (FPV) virus-like particle (VLP) vaccine and evaluated its immunogenicity.

HI antibody is considered the gold standard for FPV antibody detection [24]. HI titers of 1:40 are considered to be protective against FPV infection [25]. In this study, cats immunized with the recombinant FPV VLPs elicited a strong immune response, with HI antibody titers greater than 10log_2_ after immunization with 15 μg FPV VLPs and complete protection after challenge with FPV strain 708. Studies have reported that following immunization with inactivated FPV vaccines, the mean HI antibody titers were only 1:36, whereas the titer could exceed 1:1000 after immunization with attenuated vaccines [26]. The HI antibody titer induced by the FPV VLP vaccine in this study was comparable to that of the attenuated vaccine. The mean HI antibody titer at 3 weeks post oral immunization with CPV VP2 protein expressed in silkworm pupae was approximately 7log_2_ [13].

The collection of anticoagulated blood samples at 6 days post-challenge was based on the reported clinical course of FPV and CPV infections in cats and dogs. According to previous studies [27], typical clinical symptoms, including leukopenia (a hallmark of parvovirus infection), generally develop between 4- and 8-days post-infection. Day 6 was selected as a representative time point within this window, when viral-induced immunosuppression and hematological changes are expected to be most pronounced. By collecting samples at this time, we aimed to effectively capture the impact of infection and potentially the protective effects of vaccination on white blood cell counts. This approach is consistent with standard practices in similar experimental infection models, where day 6 post-challenge is often used for assessing immune status and disease progression.

Significant progress has also been made in the application of virus-like particle (VLP) technology in vaccine research. VLPs effectively present and deliver conformational antigens, thereby inducing the adaptive immune response (i.e., cell-mediated and humoral immunity). This study demonstrated that combining VLPs with gel adjuvant can elicit a robust immune response.

## 5. Conclusions

In this study, we developed FPV virus-like particles (VLPs) expressing the VP2 protein as a potential vaccine candidate against feline panleukopenia virus (FPV). This approach elicited a robust humoral immune response and provided complete protection following challenge with virulent FPV. The antigenic properties of the FPV VLPs demonstrate their potential for developing novel FPV vaccines, as well as chimeric VLP-based vaccines targeting other animal diseases.

All cats used in this study were initially FPV antibody less than 1:4, which may have contributed to the observed high efficacy of the VLP vaccine. Future studies should consider evaluating the immunogenicity and protective efficacy of the FPV VLP vaccine in animals with pre-existing FPV immunity, as this could influence vaccine performance in real-world scenarios.

## Figures and Tables

**Figure 1 vaccines-13-00684-f001:**
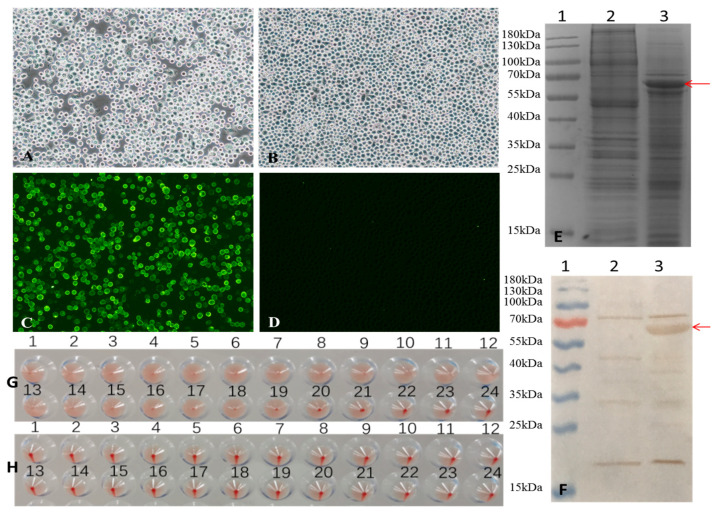
Generation of recombinant baculovirus and identification of VP2 proteins expressed in Sf-9 cells. (**A**,**B**) Infection with recombinant baculovirus alters the morphology of Sf9 cells. (**A**) Sf9 cells infected with recombinant baculovirus for 72 h, and (**B**) Sf9 control cells were observed under a light microscope (magnification, 100×). (**C**,**D**) Indirect fluorescence assay (IFA) of VP2 protein expressed in Sf9 cells. (**C**) Sf9 Cells infected with the recombinant baculovirus for 72 h and (**D**) Sf9 control cells were identified by IFA and observed under a fluorescence microscope (magnification at 100×). (**E**,**F**) SDS-PAGE and Western blot analysis of VP2 protein expression. (**E**) 12%SDS-PAGE and Coomassie Brilliant Blue staining and (**F**) Western blot analysis. Lane 1, pre-stained protein markers; lane 2, Sf9 control cells; lane 3, Sf9 cells infected with recombinant baculovirus. The red arrows indicate the FPV VP2 protein. (**G**,**H**) hemagglutination (HA) activity of FPV VP2. HA titers were detected using 1% pig erythrocytes. (**G**) Sf9 cells infected with recombinant baculovirus for 72 h, and (**H**) Sf9 control cells were observed.

**Figure 2 vaccines-13-00684-f002:**
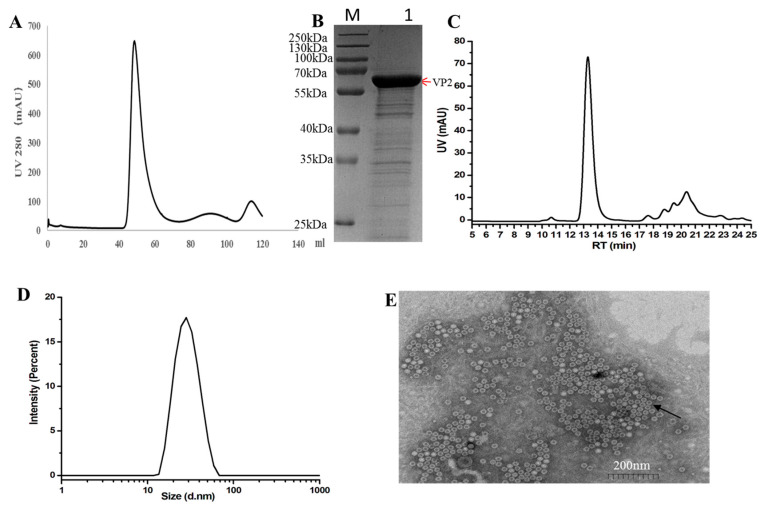
Purification and characterization of FPV virus-like particles (VLPs). (**A**) Gel filtration chromatography profile of FPV VLPs using a Hi Load 16/60 Superdex 200-pg column. (**B**) 12%SDS-PAGE and Coomassie Brilliant Blue staining analysis of proteins obtained from gel filtration chromatography. The red arrows indicate the FPV VP2 protein. (**C**) Size-exclusion chromatography (SEC) profile of purified FPV VLPs, showing the main peak corresponding to nanoparticle formation. (**D**) Particle size distribution of FPV VLPs as determined by dynamic light scattering (DLS), with an average diameter of approximately 25 nm. (**E**) Transmission electron microscopy (TEM) image of FPV VLPs, negatively stained with 3% phosphotungstic acid, showing spherical particles with uniform morphology and a diameter of approximately 25 nm. Scale bar = 200 nm. The black arrows indicate FPV VLPs.

**Figure 3 vaccines-13-00684-f003:**
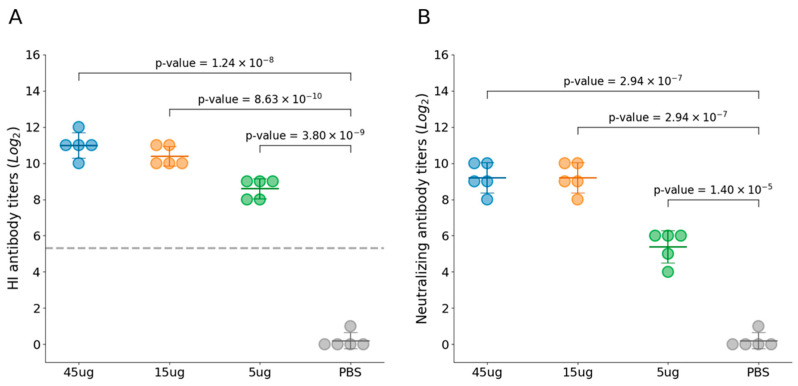
Immunogenicity induced by FPV VLPs. (**A**) The hemagglutination inhibition (HI) titer in the serum of cats from Groups 1 to 4 was determined using HI assay with 1% pig erythrocytes. (**B**) Virus-neutralizing antibody titers were measured by IFA and expressed as the reciprocal of the highest dilution at which infection of the F81 cells was inhibited in 50% of the culture wells. Blue represents Group 1 (45 μg VLPs immunization), yellow represents Group 2 (15 μg VLPs), green represents Group 3 (5 μg VLPs), and gray represents Group 4 (PBS control). All groups contained an equal number of cats (*n* = 5 per group). The horizontal lines represent the group means, the error bars indicate the standard deviation, and the gray dashed line represents the 1:40 HI antibody titer.

**Figure 4 vaccines-13-00684-f004:**
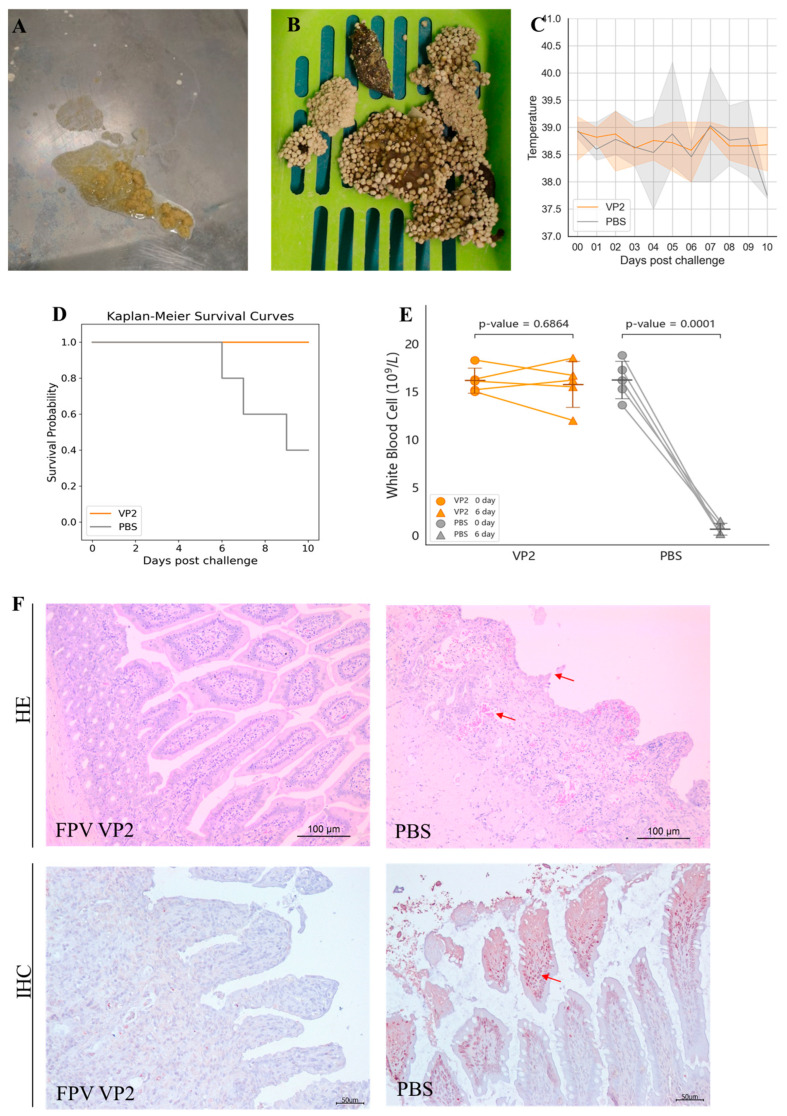
Protective efficacy of FPV VLPs. (**A**,**B**) Severe vomiting and diarrhea were observed in the PBS control group after challenge. (**C**) Body temperature monitoring after challenge, showing the range of maximum and minimum values within the group. (**D**) Survival rate after viral challenge. (**E**) White blood cell count on day 6 post-challenge. (**F**) Intestinal tissues (jejunum) were analyzed by hematoxylin and eosin (HE) staining and immunohistochemistry (IHC) assays. The arrows indicate villous atrophy and shortening of the jejunum, as well as hemorrhage/congestion in H&E staining; The arrows point to FPV positivity in IHC.

**Table 1 vaccines-13-00684-t001:** Experimental design of VLP-based FPV immune efficacy studies.

Group	Number of Cats	Immunogen (1.0 mL/dose)	Dose
1	5 (1#-5#)	VLPs (45 μg)	1.0
2	5 (6#-10#)	VLPs (15 μg)	1.0
3	5 (11#-15#)	VLPs (5 μg)	1.0
4	5 (16#-20#)	PBS	1.0

**Table 2 vaccines-13-00684-t002:** Clinical symptoms during challenge in VLP Group 2 and Control Group 4.

	Loss of Appetite ^a^	Depression ^b^	Vomiting ^c^	Diarrhea ^d^	Fever ^e^
Group 2	0/5	0/5	0/5	0/5	0/5
Group 4	5/5	5/5	2/5	5/5	3/5
*p*-value (Fisher’s exact test)	0.0079	0.0079	0.4444	0.0079	0.1667

^a^ Number of cats showing loss of appetite/total number of cats in the group. ^b^ Number of cats showing depression/Total number of cats in the group. ^c^ Number of cats showing vomiting/Total number of cats in the group. ^d^ Number of cats showing diarrhea/Total number of cats in the group. ^e^ Number of cats showing fever (body temperature exceeding 39.5 °C)/total number of cats in the group.

## Data Availability

Data will be made available upon request.

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
