# Peer review of "Virus-like Particle Vaccine for Feline Panleukopenia: Immunogenicity and Protective Efficacy in Cats"

_vaccines, 2025, doi:10.3390/vaccines13070684_

Round 1
Reviewer 1 Report
Comments and Suggestions for Authors
The article "Virus-Like Particle Vaccine for Feline Panleukopenia: Immunogenicity and Protective Efficacy in Cats" by Wang et al. aimes to develop a new VLP-based vaccine against feline panleukopenia in cats. As part of the study, the authors obtained the feline panleukopenia virus VP2 protein using a baculovirus expression system, characterized it in detail, and used it to immunize laboratory animals (specifically, 20 cats). An important and positive aspect of this study was the confirmation of virus-like particle formation via transmission electron microscopy. The results demonstrate the induction of strong protection against the virus following immunization with various VLP doses. Undoubtedly, these findings represent a first step toward developing a new vaccine. However, the experimental design was limited due to the small number of animals. Also several important controls were missing, such as VLP alone, VLP with alternative adjuvants, and comparisons with licensed vaccines. Nevertheless, the data suggest that this approach holds promise for the development of a new effective and safe vaccine.
Below are several questions and recommendations that, in my view, could significantly enhance the quality of the article.
Line 38 When describing the disease, I recommend adding information on mortality rates. The high risk of lethality is often a key driver for vaccine development.
Line 54 This section provides general information about VLP-based vaccines. However, parvoviruses and their VLPs do not exhibit rod-like structures. I think that this text could mislead readers.
Line 59 According to the ICTV database, the species Feline parvovirus was abolished in 2005. As of the 2024 release, it is now classified as Protoparvovirus carnivoran 1. Please include the current taxonomic name (can be given in parentheses).
Line 119 Please specify the DLS instrument model, the grid preparation protocol (or a reference), the electron microscope model, and the staining method used.
Line 124 Why was the Seppic adjuvant selected for this study? Since the experiments lack VLP-alone control groups, the adjuvant’s contribution remains unclear. Do the authors have preliminary data or rationale for this choice, which should be included in the Discussion? Moreover, the adjuvant is not mentioned in the Discussion section, which could lead to incorrect assumptions about immune response induction by VLPs alone.
Line 156 Clarify how data are presented in the graphs (medians, means, error bars).
Line 166–167 Likely, "control" should be added before "Sf9 cells." Please verify.
Line 172 What is the predicted molecular weight of the protein? (For review only.)
Line 177 Do you mean "control cells" in the last sentence?
Line 178 Explain the presence of additional bands in Figure 1D (lanes 2 and 3).
Line 178 Figure 2B demonstrates significant additional peaks on the right. Include information about purity and protein percentage in the Results section.
Line 196 Provide the standard deviation values for VLP diameter calculated from TEM image analysis.
Line 185 In the Figure caption, specify the percentage of the SDS-PAGE gel and the staining method used.
Line 201 In the Figure caption, clarify the staining method for VLPs (uranyl acetate?).
Line 209 I recommend including the raw immunogenicity assay data as supplementary materials.
Line 211 In the Figure caption, specify the type of error bars and the post-hoc test used. This information should also be added to the Materials and Methods section.
Line 224 Provide details (or a reference) on the methodology for white blood cell measurement and data processing. What was considered as 100%?
Line 225 Which statistical test was applied here? Add this information to the Materials and Methods section.
Line 228 The Discussion does not provide a thorough comparison with current vaccines and potential advantages of this vaccine candidate in terms of efficacy and safety.
Author Response
We sincerely appreciate the reviewers’ insightful comments and constructive suggestions, which have significantly improved the quality of our manuscript. Below are our point-by-point responses to the reviewers’ concerns.
Please see the attachment.

Reviewer 2 Report
Comments and Suggestions for Authors
Wang et al, have assessed the immunogenicity of a virus-like particle vaccine candidate against feline panleukopenia virus (FPV). The authors produced this novel vaccine candidate using baculovirus-driven insect cell expression, leading to the production of VP2-derived FPV VLPs, which were characterised by dynamic light scattering and electron microscopy. Wang et al, then immunised cats at different doses and assessed the immunogenicity through a challenge study, using a virulent strain of FPV. The authors found that immunisation with the VLPs in the presence of adjuvant led to a robust HI-titre, whilst also observing protection from severe clinical disease presentation. I believe this paper to be of interest to the community with some positive results, however, I feel that the data presented also raises a number of questions which will need to be addressed before it can be accepted for publication.
Major Comments
Line 176-177 – The authors have described HA titre data for the VP2 VLPs but have not shown the data for this in the manuscript. The authors should present this data in a figure or remove this statement from the manuscript.
Section 3.3 – The authors describe the immunogenicity of the FPV VLPs, however I have some questions regarding this section that would need to be addressed prior to publication:
- The immunogenicity assay requires some description in the results section to orientate the reader rather than having to go back to the methods section to understand how these experiments were performed.
- What is the rationale for the amount of protein used in each immunisation group, 45 micrograms of protein seems a larger dose, how does this compare to the protein concentration of inactivated vaccine dosing?
- Figure 3 – The axis labelling is inconsistent between the two graphs with bold brackets. Additionally, the figure legend does not describe these experiments in any detail. The figure legend should reflect the experiment in its entirety.
- Could the authors address why they chose not to assess the VLP vaccine candidate in comparison to the currently licensed inactivated vaccine?
- Line 212 – The authors describe the clinical severity scores but do not show this data in the form of a chart or supplementary data which display the clinical severity across the duration of the challenge study.
- Figure 4 – 4A is comparing the number of leukocytes to a baseline at 6 days post-infection. Can the authors show the baseline data? Such as was this done individually for each cat pre- and post-vaccination and challenge and why did the authors choose 6 days post-challenge?
- Additionally, the organisation of Figure 4 isn’t in the correct order as 4B and 4C appear to the left of 4A
Discussion – The authors do not really discuss the advantages of using the baculovirus system over other expression systems, including mammalian, particularly in the context of cost, time and ease. Further to this, the authors do not necessarily compare the potential of this vaccine candidate to that of the current vaccines available. They also conclude that a single dose of FPV VLPs can induce HI titres to the gold standard, however they should also highlight that this is in the presence of adjuvant and maybe down to the high amount of protein used to immunise with.
Minor Comments
Line 47 – The authors refer to inactivated vaccines requiring adjuvants and multiple doses to induce immunity, however, multiple doses isn’t specific to inactivated vaccines, as highlighted by human vaccines for poliovirus and measles virus, as well as feline calicivirus, which also requires a multiple dose-regimen despite being an attenuated vaccine. Further to this, not all inactivated vaccines require adjuvants, as IPV is capable of inducing long-term, protective immunity without an adjuvant. Therefore, I believe the authors should alter these sentences to more accurately reflect the pros and cons of these different types of vaccine.
Line 52-58 – The authors could provide further references and examples describing VLP vaccine candidates in development or licensed VLP vaccines currently in use within the veterinarian or human immunisation programme, such as Foot-and-mouth-disease virus.
Line 59-65 – The authors could provide further details of parvovirus structure and the roles of the individual capsid proteins including VP1 (& VP3) alongside VP2, this would provide increased context to the readership.
Line 105-106 – The HA assay should be described briefly, even if referenced to a previous publication.
Line 122 – This sentence doesn’t fit in the methods section, therefore if the authors wish to discuss this, they should add this into the discussion and elaborate further.
Figure 1 – The labelling of Figure 1 is confusing, I suggest labelling each image individually, so that the naming runs from 1a-1f.
Author Response

(The authors gave the same response as above.)

Reviewer 3 Report
Comments and Suggestions for Authors
The aim of this paper is to create virus-like particles of the VP2 protein of feline panleucopenia virus (FPV) and to determine their immunogenic and protective effects in a vaccine trial. The rationale is that although FPV vaccines are highly effective, particularly the widely used modified live vaccines, there is a possibility of reversion to virulence. This has very occasionally occurred in the past, but in reality this is highly unlikely given the safety tests required by the licensing authorities.
The paper is generally well written and presented and the laboratory and molecular methodology is clearly described. I note that the experimental cat challenge trial was approved by the Animal Care and Ethics Committee at the authors’ institution but I have concerns about the following issues.
- What was the source of the cats – were they bred specifically for experimental studies, and were they specific pathogen free? The age of the cats was said to be 20 weeks – no breed or sex was given though the cats were divided randomly into the four groups. There is no reference in the main body of the paper that the cats have been screened for FPV antibodies prior to vaccination, though the unvaccinated controls did appear to be seronegative throughout the study period (but see below). And then – somewhat surprisingly in the conclusion - the authors do state that the cats used in the study were all antibody negative, but no details of the methodology or timing (immediately prior to vaccination?) are given.
However lines 205 – 208 and Figures 3A and 3B raise some concerns. The words ‘In general’ re the vaccinated cats having significantly higher HI and VN antibody titres compared to controls 21 days post vaccination, suggest that despite the statistics, some of the control cats may have had some previous exposure. Interestingly the HI and VN antibody titres in the vaccinated group seem to be relatively high for a VLP response without any prior exposure to the virus, though this might be possible.
- FPV strain 708 was used both to create the virus like particles, and was also the challenge virus. The authors state that this was obtained from a sick cat and was said to be virulent. Although this does appear to be the case, and although it is unlikely, it is important for the challenge trial that the authors clarify that the virus was eg plaque purified or checked for potential contaminant viruses that might also play a role alongside FPV re the clinical signs following challenge.
- Given the unvaccinated and challenged cats showed what was said to be severe clinical signs including vomiting and diarrhoea, what were the criteria for allowing the experiment to continue until 10 day when (and how?) they were killed. Most ethical committees and licensing authorities would expect some early intervention – and possibly euthanasia- in the trial if serious suffering was occurring.
- Generally in FPV vaccine challenge trials it can be difficult to invoke severe clinical signs and the criteria used for efficacy may rely on a white blood cell count, and milder signs such as fever, and some depression and loss of appetite. Given the unvaccinated challenged cats were severely ill, could the authors confirm whether or not they withheld food eg for 24 hours before challenge, which can enhance clinical signs because of increased replication of intestinal epithelial cells targeted by FPV.
- Other minor points. Line 35. Feline plague not used nowadays. Line 127. Blood samples were collected. Line 177. Sf9 control cells.
Author Response

(The authors gave the same response as above.)

Round 2
Reviewer 1 Report
Comments and Suggestions for Authors
The authors responded to all of the reviewer's comments and made the necessary changes to the manuscript text. Significant changes include additional information about research protocols and methods of statistical processing of results.
Author Response
The manuscript has been revised according to the editor's comments.